# Synthesis, Crystal Structure and Photoluminescent Properties of Red-Emitting CaAl$_4$O$_7$:Cr$^{3+}$ Nanocrystalline Phosphor

**Leonid Vasylechko** [1,*], **Vitalii Stadnik** [2], **Vasyl Hreb** [1], **Yaroslav Zhydachevskyy** [3], **Andriy Luchechko** [4], **Vitaliy Mykhaylyk** [5], **Hans Kraus** [6] and **Andrzej Suchocki** [3]

1. Semiconductor Electronics Department, Lviv Polytechnic National University, 12 Bandera Str., 79013 Lviv, Ukraine; vasyl.m.hreb@lpnu.ua
2. Department of Physical, Analytical and General Chemistry, Lviv Polytechnic National University, 12 Bandera Str., 79013 Lviv, Ukraine; vitalii.y.stadnik@lpnu.ua
3. Institute of Physics, Polish Academy of Sciences, Al. Lotników 32/46, 02-668 Warsaw, Poland; zhydach@ifpan.edu.pl (Y.Z.); suchy@ifpan.edu.pl (A.S.)
4. Department of Sensor and Semiconductor Electronics, Ivan Franko National University of Lviv, Tarnavskogo Str., 107, 79017 Lviv, Ukraine; andriy.luchechko@lnu.edu.ua
5. Diamond Light Source, Harwell Campus, Didcot OX11 0DE, UK; vitaliy.mykhaylyk@diamond.ac.uk
6. Physics Department, University of Oxford, Keble Rd., Oxford OX13 3RH, UK; hans.kraus@physics.ox.ac.uk
* Correspondence: leonid.o.vasylechko@lpnu.ua; Tel.: +38-096-667-6279

**Abstract:** Calcium dialuminate, CaAl$_4$O$_7$, nanopowders with a grossite-type structure, doped with chromium ions, were synthesized via the combined sol–gel solution combustion method. The evolution of phase composition, crystal structure, and microstructural parameters of the nanocrystalline materials depending on the temperature of the thermal treatment was investigated via X-ray powder diffraction and applying the Rietveld refinement technique. The photoluminescent properties of CaAl$_4$O$_7$ nanophosphors activated with Cr$^{3+}$ ions were studied over the temperature range of 4.5–325 K. The samples show deep red and near-infrared luminescence due to the $^2E \rightarrow {}^4A_2$ and $^4T_2 \rightarrow {}^4A_2$ energy level transitions of Cr$^{3+}$ ions under excitation in the two broad emission bands in the visible spectral region. The R lines emission reveals a strong temperature dependence. The feasibility of the material for non-contact luminescence sensing is investigated, and good sensitivity is obtained based on the (R2/R1) luminescence intensity ratio and the lifetime of the emission.

**Keywords:** calcium dialuminate; Cr-doping; combustion synthesis; crystal structure; luminescence; non-contact temperature sensing

## 1. Introduction

Calcium dialuminate, CaAl$_4$O$_7$ (grossite), with a monoclinic $C2/c$ structure has been intensively studied during recent decades as a promising rare-earth (RE)-free host material for diverse luminescent applications [1–10]. CaAl$_4$O$_7$-based phosphors doped with rare-earth elements (Nd, Sm, Eu, and Yb) and bismuth have attracted significant attention due to their high quantum efficiency, long persistence of phosphorescence, and suitable emitting color. In particular, Sm$^{3+}$-doped CaAl$_4$O$_7$ yielding an intense orange-red emission under violet excitation [1] was proposed for lighting and display applications [2]. Calcium aluminate powders co-doped with Yb$^{3+}$ and Eu$^{3+}$ exhibit the near-infrared-to-red up-conversion luminescence of Eu$^{3+}$ [3,4]. The luminescence of Bi$^{3+}$ ions in a CaAl$_4$O$_7$ matrix is suitable for improving the rendering index of white LEDs, whereas the emission of Bi$^{2+}$ at 720–850 nm appearing in the 1st biological window is attractive for bioimaging applications [5]. Mn$^{4+}$- or Cr$^{3+}$-doped materials are considered as alternatives to RE-activated phosphors for use as cheap near-IR and deep-red emitting phosphors (see [6] and the references herein). As expected, a number of studies on Mn$^{4+}$-doped CaAl$_4$O$_7$-based luminophores emerged during the last decade [7–10]. In particular, just recently, a CaAl$_{12}$O$_{19}$–CaAl$_4$O$_7$–MgAl$_2$O$_4$:Mn$^{4+}$

(CCM:$Mn^{4+}$) red-emitting phosphor with excellent quantum efficiency and superior color purity was reported [10]. There, the great potential for using CCM:$Mn^{4+}$ phosphors for plant growth LEDs and white light-emitting diode (WLED) applications was demonstrated. However, to the best of our knowledge, no information is available in the scientific literature on chromium-doped $CaAl_4O_7$ phosphors. To fill this gap, we conducted this study that encompasses the precise structural characterization of $Cr^{3+}$-doped $CaAl_4O_7$ nanopowders synthesized via a facile sol–gel combustion method and the study of their photoluminescent properties with temperature, aiming to develop new deep-red emitting phosphors that are potentially suitable for non-contact luminescence thermometry.

## 2. Results and Discussion

### 2.1. Phase Composition, Crystal Structure, and Microstructural Parameters

The X-ray diffraction (XRD) of the product of nominal composition $CaAl_{3.9995}Cr_{0.0005}O_7$ obtained after a spontaneous auto-combustion process at 800 °C (see Section 3.1) revealed the formation of an X-ray amorphous material (Figure 1, top panel).

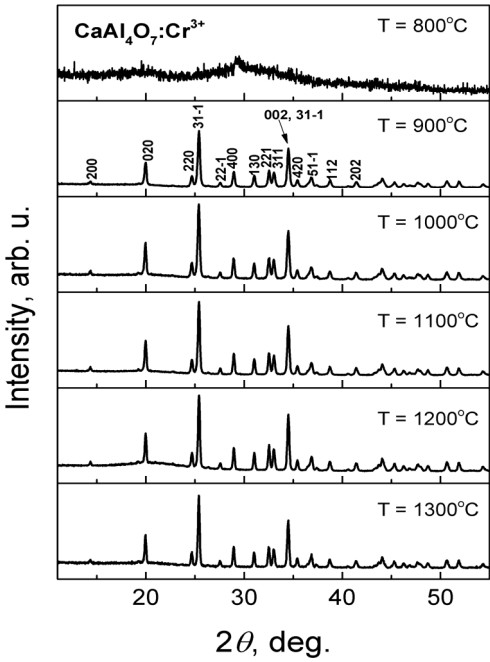

**Figure 1.** Evolution of XRD patterns of $CaAl_4O_7$:$Cr^{3+}$ powders, which were successively annealed at different temperatures. Miller's indices for the monoclinic $CaAl_4O_7$ structure are given on the T = 900 °C diffractogram.

The crystallization of $CaAl_4O_7$ and formation of a phase-pure grossite-type material with an average grain size $<D_{ave}>$ of 76 nm occurs after annealing the powder at 900 °C. The grain size was evaluated from the angular dependence of the shape of Bragg's maxima via full-profile Rietveld refinement. Further heat treatments of the powder at 1000, 1100, 1200, and 1300 °C do not impact on the phase composition of the material, but they significantly improve the crystallinity of the sample. This is reflected by a gradual decrease of the lattice strains $<\varepsilon>$ in $CaAl_4O_7$:$Cr^{3+}$ structure from 0.157% to 0.113% and the simultaneous increase in the average grain size of up to 166–198 nm (see Figure 2a). The last tendency was also confirmed via scanning electron microscopy (SEM) examinations of the powders heat treated at different temperatures. The obtained SEM images (Figure 2b–d) confirm an essential increase in the sizes of the weakly faceted and strongly agglomerated grains from about 100 nm for the powder calcined at 1000 °C up to ≥200 nm for the powder calcined at 1300 °C.

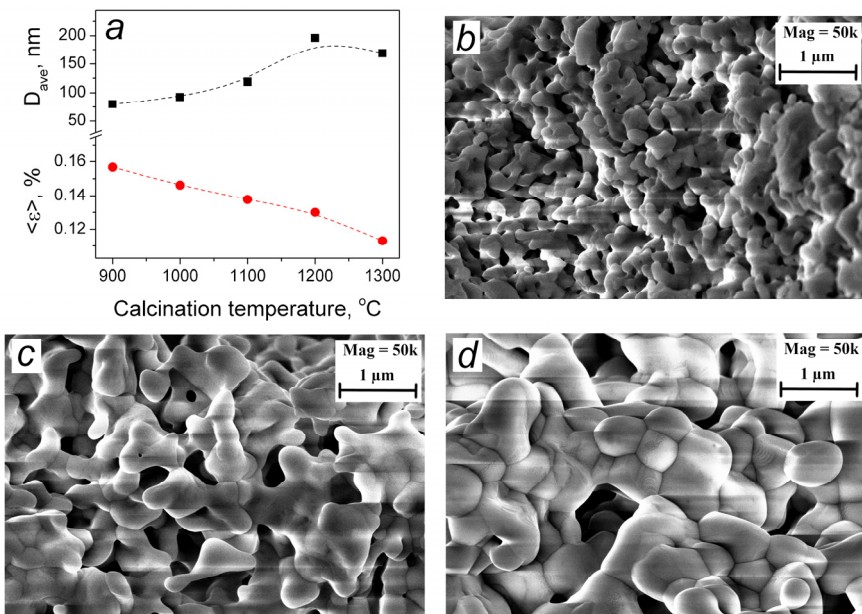

**Figure 2.** (**a**) Lattice strain values <ε> and average crystallite size $D_{ave}$ of $CaAl_4O_7:Cr^{3+}$ powders, depending on the annealing temperature. Dashed B-Spline lines are provided as a guide to the eye. (**b**–**d**) SEM images of the studied $CaAl_4O_7:Cr^{3+}$ powder samples calcined at 1000 °C (**a**), 1200 °C (**b**), and 1300 °C (**c**).

Full-profile Rietveld refinement was also conducted for the evaluation of precise structural parameters of the $CaAl_4O_7:Cr^{3+}$ series. As the initial model for the refinement, atomic coordinates in a nominally pure $CaAl_4O_7$ ($CaO·2Al_2O_3$) compound derived from single crystal diffraction data in Ref. [11] were used. In the refinement procedure, the unit cell dimensions, coordinates, and displacement parameters of metal and oxygen atoms were refined together with the profile parameters and corrections for absorption and the instrumental sample shift. In all cases, we achieved an excellent agreement between the calculated and experimental diffraction profiles. Example results of the Rietveld refinement of $CaAl_4O_7:Cr^{3+}$ structures are presented in Figure 3. Obtained structural parameters for $CaAl_4O_7:Cr^{3+}$ powders annealed at 1200 °C and the corresponding interatomic distances calculated from these data are presented in Tables 1 and 2, respectively.

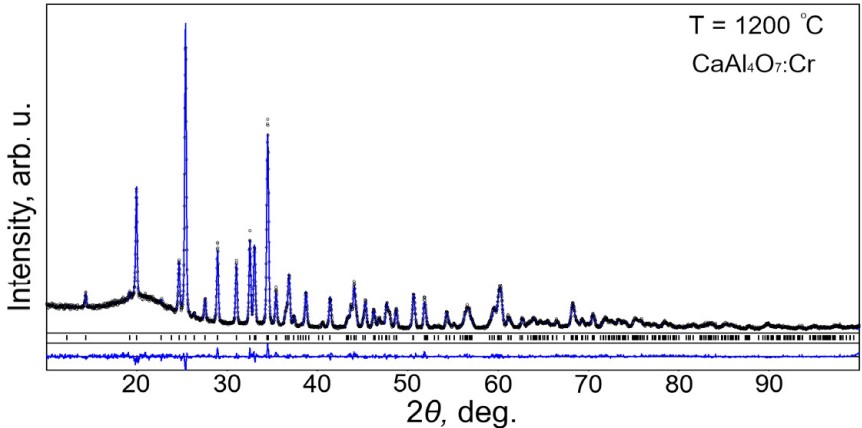

**Figure 3.** Experimental X-ray diffractogram of $CaAl_4O_7:Cr^{3+}$ powder annealed at 1200 °C (black circles) in comparison with calculated XRD pattern (blue lines). The difference curve between the two graphs is shown at the bottom. Vertical bars pointing to Bragg's maxima indicate the positions in the monoclinic grossite-type structure.

**Table 1.** Lattice parameters, coordinates and displacement parameters of atoms in a $CaAl_4O_7:Cr^{3+}$ structure (SG $C2/c$, Z = 4) after heat treatment at 1200 °C.

| Lattice Parameters, Residuals | Atom, Sites | x/a | y/b | z/c | $B_{iso/eq}$, Å² |
|---|---|---|---|---|---|
| | $CaAl_4O_7:Cr^{3+}$ @ 1200 °C; $R_I$ = 0.038, $R_P$ = 0.109 | | | | |
| | Ca, 4e | 0 | 0.8102(3) | 1/4 | 1.02(6) |
| $a$ = 12.8947(6) Å | Al1, 8f | 0.1634(2) | 0.0864(3) | 0.3036(6) | 0.95(6) |
| $b$ = 8.8876(4) Å | Al2, 8f | 0.1204(2) | 0.4420(3) | 0.2427(5) | 0.78(6) |
| $c$ = 5.4420(3) Å | O1, 4e | 0 | 0.5275(7) | 1/4 | 1.0(2) |
| $\beta$ = 107.017(1)° | O2, 8f | 0.1164(4) | 0.0543(5) | 0.5669(9) | 0.73(12) |
| $V$ = 596.36(9) Å³ | O3, 8f | 0.1213(3) | 0.2553(5) | 0.1551(10) | 0.88(11) |
| | O4, 8f | 0.1931(4) | 0.4400(5) | 0.5836(10) | 0.85(12) |

**Table 2.** The nearest interatomic distances (up to 4.0 Å) in $CaAl_4O_7:Cr^{3+}$ sample annealed at 1200 °C.

| Atoms | Distances [Å] | Atoms | Distances [Å] | Atoms | Distances [Å] |
|---|---|---|---|---|---|
| | | $CaAl_4O_7:Cr^{3+}$ @ 1200 °C | | | |
| Ca-O3 ×2 | 2.361(5) | Al1-O3 | 1.716(5) | Al2-O3 | 1.727(5) |
| Ca-O2 ×2 | 2.372(5) | Al1-O2 | 1.735(5) | Al2-O1 | 1.739(4) |
| Ca-O1 | 2.512(6) | Al1-O2 | 1.770(5) | Al2-O4 | 1.790(5) |
| Ca-O2 ×2 | 2.902(5) | Al1-O4 | 1.786(5) | Al2-O4 | 1.818(6) |
| Ca-Al1 ×2 | 3.193(3) | Al1-Al1 ×2 | 3.125(4) | Al2-O1 | 2.690(3) |
| Ca-Al1 ×2 | 3.263(3) | Al1-Al2 | 3.146(4) | Al2-Al2 ×2 | 2.910(4) |
| Ca-Al2 ×2 | 3.491(3) | Al1-Al2 | 3.147(4) | Al2-Al2 | 3.128(4) |
| Ca-Al2 ×2 | 3.627(4) | Al1-Al2 | 3.208(4) | Al2-O2 | 3.246(5) |
| Ca-O4 ×2 | 3.644(5) | Al1-O4 | 3.441(5) | Al2-O3 | 3.501(5) |
| Ca-Al1 ×2 | 3.766(3) | Al1-O4 | 3.464(5) | Al2-O1 | 3.551(3) |

Aluminum atoms in the $CaAl_4O_7$ structure occupy two different general 8f positions and are characterized by a distorted tetrahedral environment of oxygen atoms at distances of 1.72–1.79 Å and 1.73–1.82 Å away from the central atoms for the positions Al(1) and Al(2), respectively (Table 2, Figure 4). What is peculiar about the Al(2) position is the presence of five remote oxygen atoms located at 2.69 Å away from a central Al atom; there are three Al atoms located at the distances of 2.91–3.13 Å (see Table 2). In contrast, in the Al(1) site, the second coordination environment is formed by five Al atoms located at 3.13–3.21 Å; there are no extra atoms between the first tetrahedral and second coordination spheres.

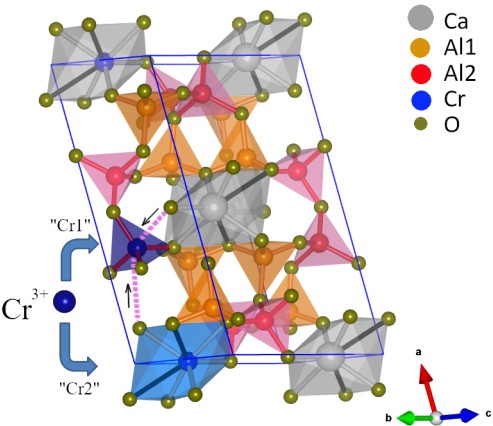

**Figure 4.** Polyhedral view of the $CaAl_4O_7$ structure as a framework of two types of vertices-shared $AlO_4$ tetrahedra linked with $CaO_7$ pentagonal bipyramids. Two possible sites for incorporation of $Cr^{3+}$ ions into $CaAl_4O_7$ lattice are shown.

Calcium atoms in the $CaAl_4O_7$ structure are located at the centers of trigonal bipyramids formed by five oxygens located at distances of 2.36–2.51 Å away from the central atoms (Table 2). Two remote oxygen atoms located at distances of 2.89 Å could also be assigned to the nearest coordination sphere of Ca atoms. In this case, the coordination number (CN) of Ca becomes seven, and the corresponding polyhedron is shaped as a pentagonal bipyramid, which is similar to Sr atoms in the related $SrAl_4O_7$ compound [12,13]. Thus, the crystal structure of $CaAl_4O_7$ can be seen as a 3D grid of $CaO_7$ polyhedra connected with two kinds of apex-shared $AlO_4$ tetrahedra (Figure 4). As it was shown in earlier work on $CaAl_4O_7$ and $SrAl_4O_7$ structures [14], due to the 7:4 oxygen-to-aluminum ratio in this structure type, one oxygen atom (O4) is bonded to three aluminum atoms, thus connecting three $AlO_4$ tetrahedra via a common corner. To obtain a complete network of $AlO_4$ tetrahedra, in which each oxygen atom is bonded to only two aluminum atoms, a 2:1 ratio would be required.

The detailed analysis of the obtained structural parameters of materials heat treated at different temperatures revealed minor, but detectable anisotropic behaviors of the lattice parameters of the $CaAl_4O_7$:$Cr^{3+}$ structure. The lattice parameters $a$ and $b$, as well as the monoclinic angle $\beta$ slightly decrease with an increasing annealing temperature, while the $c$ parameter increases (see Figure 5). As a result, the unit cell volume is nearly independent of the heat treatment temperature (see Figure 5, bottom panel).

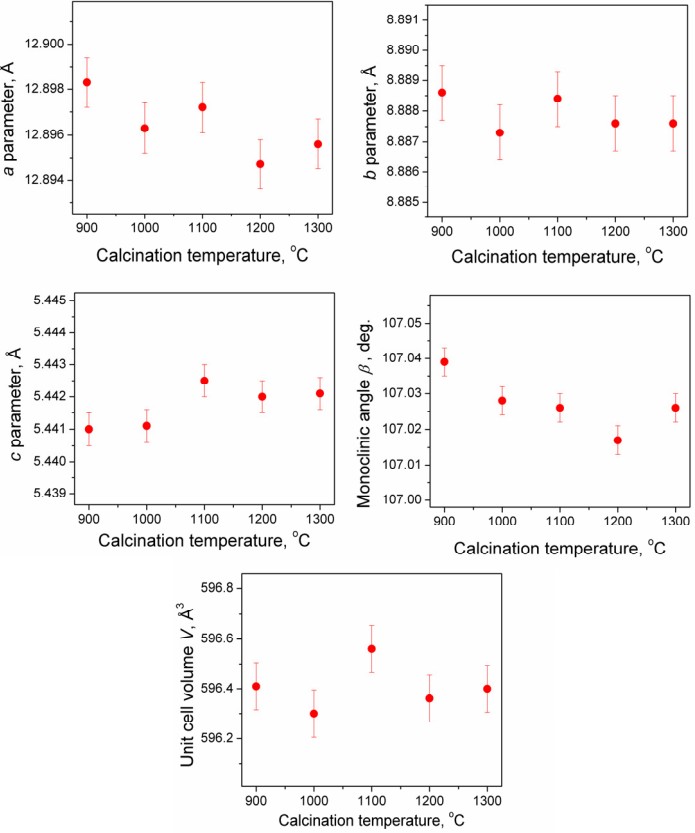

**Figure 5.** Dependencies of the lattice parameters, $a$, $b$, and $c$, monoclinic angle, $\beta$, and unit cell volume, $V$, of $CaAl_4O_7$:$Cr^{3+}$ on the annealing temperature of the powders.

## 2.2. Luminescent Properties

Photoluminescence excitation (PLE) and corresponding photoluminescence (PL) spectra of $CaAl_4O_7$: $Cr^{3+}$ nanocrystalline phosphors annealed at 1200 °C are shown in Figure 6. The sample calcined at 1200 °C reveals two types of $Cr^{3+}$ emission. The first one has narrow lines (R lines) at 692.5 and 693.9 nm at room temperature, which are undoubtedly caused by $^2E \rightarrow {}^4A_2$ spin-forbidden transitions in the $Cr^{3+}$ ion in a strong octahedral crystal field

(O-Cr-A type, according to Refs. [15,16]). The inset in Figure 6b shows the R lines of this center in comparison with a similar center previously observed by us in $SrAl_4O_7:Cr^{3+}$ [13]. The second one also has two narrow lines at 683.3 and 686.7 nm, which are superimposed on a broad emission band stretching up to 800 nm. The excitation bands caused by $^4A_2 \rightarrow {}^4T_1$ and $^4A_2 \rightarrow {}^4T_2$ transitions in the octahedrally coordinated $Cr^{3+}$ ion for the second-type center are somewhat shifted towards longer wavelengths (see Figure 6a). The broad emission band of the second-type center also caused by spin-allowed $^4T_2 \rightarrow {}^4A_2$ transitions points towards a lower crystal field strength experienced by this type of $Cr^{3+}$ ions in the studied material (O-Cr-B type, according to Refs. [15,16]).

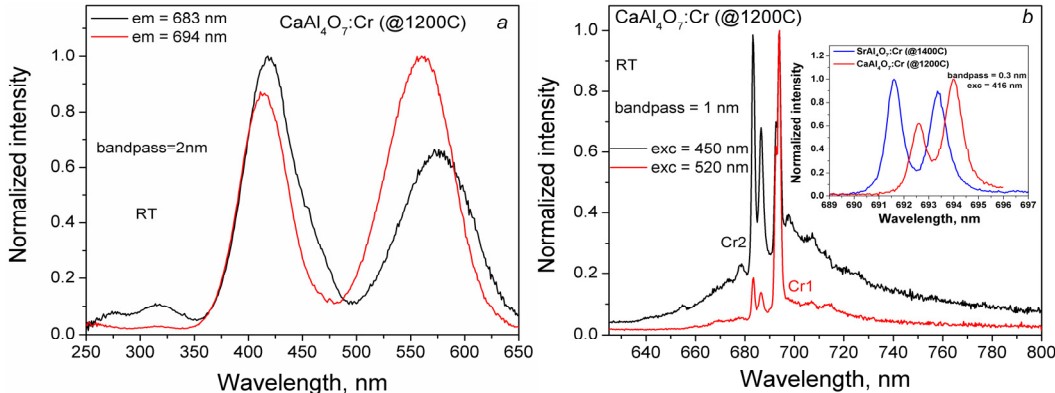

**Figure 6.** The room temperature PLE (**a**) and photoluminescence (**b**) spectra of $CaAl_4O_7:Cr^{3+}$ nanocrystalline powders calcined at 1200 °C. The inset in (**b**) demonstrates the R lines of Cr1 center at T = 300 K in $CaAl_4O_7:Cr^{3+}$ in comparison with those of $SrAl_4O_7:Cr^{3+}$ studied in [13] (red and blue lines, respectively).

It should be mentioned that the $Cr^{3+}$ center in a weak crystal field exhibiting a broadband emission was not discussed in our recent study of $SrAl_4O_7:Cr^{3+}$ [13]. Having observed this new center in the studied calcium aluminate, we revisited the earlier results and carried out additional measurements for $SrAl_4O_7:Cr^{3+}$ samples calcined at 1400 °C. The examinations confirmed that the Cr center in a weak crystal field does exist in this host (see Figure 7). This center marked as Cr2 in $SrAl_4O_7:Cr^{3+}$ manifests a broad emission band centered at about 780 nm, with excitation bands at about 470 and 630 nm.

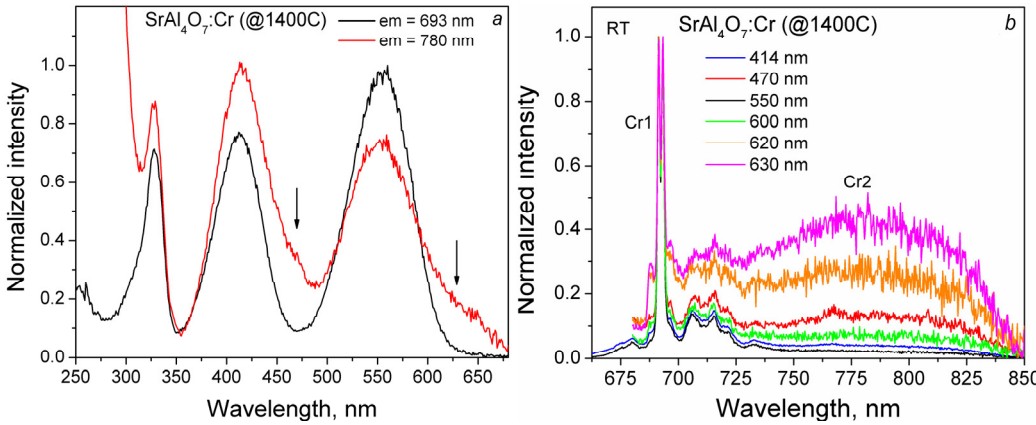

**Figure 7.** The room temperature PLE (**a**) and PL (**b**) spectra of $SrAl_4O_7:Cr^{3+}$ nanocrystalline powder calcined at 1400 °C.

Judging from the presented photoluminescence spectra of $SrAl_4O_7:Cr^{3+}$ and $CaAl_4O_7:Cr^{3+}$ nanocrystalline phosphors, one can expect that the observed two types of non-equivalent octahedrally coordinated $Cr^{3+}$ centers (in the stronger (Cr1) and weaker (Cr2) crystal fields) are

formed on the sites of $Al^{3+}$ and $Ca^{2+}/Sr^{2+}$ ions, respectively. A significant shift in the emission and excitation bands of the Cr2 center towards near-infrared with respect to the Cr1 center in $SrAl_4O_7$ compared to those of $CaAl_4O_7$ suggests that in the $SrAl_4O_7$ host, the Cr2 center experiences much weaker crystal field strength than it does in $CaAl_4O_7$. Taking into account the fact that the ionic radius of $Sr^{2+}$ cation is larger than that of $Ca^{2+}$, it is sensibly to anticipate that the $Cr^{3+}$ center (Cr2) in a weak crystal field observed both in $SrAl_4O_7$ and $CaAl_4O_7$ most likely is localized at the sites of the alkaline earth cations. However, it is not obvious given the crystal structure of the $CaAl_4O_7$ lattice described above. The origin of the $Cr^{3+}$ centers in $(Ca/Sr)Al_4O_7$ is discussed below.

Figure 8 demonstrates an evolution of the Cr-related emission of the studied material as a function of the annealing temperature. As it is clearly seen from the figure, the samples calcined at 900–1000 °C show a structureless broad-band emission at around 700 nm of relatively low intensity, which was caused by the poor crystallinity of the material. Narrow emission lines appear in the powder treated at 1100 °C. The material calcined at 1200 °C shows high-intensity photoluminescence, as described above. It is of interest to note that further increasing the annealing temperature up to 1300 °C results in a complete disappearance of the Cr2 centers, so that only the Cr1 centers remain without further increasing the emission intensity. Therefore, the material calcined at 1200 °C with two types of Cr centers was chosen to be studied in detail in the temperature range from about 4 to 325 K.

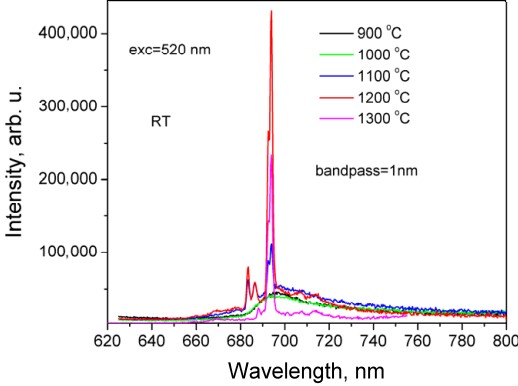

**Figure 8.** The room temperature photoluminescence spectra of $CaAl_4O_7$:$Cr^{3+}$ nanocrystalline powders at different calcination temperatures.

Figure 9 shows the detailed temperature dependence of the PL spectra in the vicinity of R lines of $CaAl_4O_7$:$Cr^{3+}$ calcined at 1200 °C. The corresponding temperature dependences of the $R_2/R_1$ intensity ratios and R lines positions for the Cr1 and Cr2 centers are analyzed in the next section.

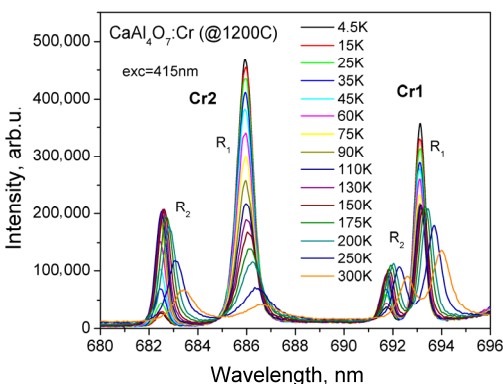

**Figure 9.** The temperature dependence of *R* line emission spectra of $CaAl_4O_7$:$Cr^{3+}$ calcined at 1200 °C.

The decay curves of both types of $Cr^{3+}$ luminescence centers in the temperature range from 4.5 to 325 K are shown in Figure 10. As can be seen from the figures, the Cr2 center emission registered at 686 nm exhibits a single instance of exponential decay. Cr1 emission decays are essentially non-single exponential ones caused by spectral overlap with the Cr2 emission spectrum. To estimate the characteristic decay times of the observed emissions, we first derived the decay time constant of the Cr2 center. Next, the decay time of the Cr2 center was used as a fixed parameter to fit the decay curves of the Cr1 emission by a sum of two exponential functions. In this way, the decay time of the Cr1 centers was reliably estimated.

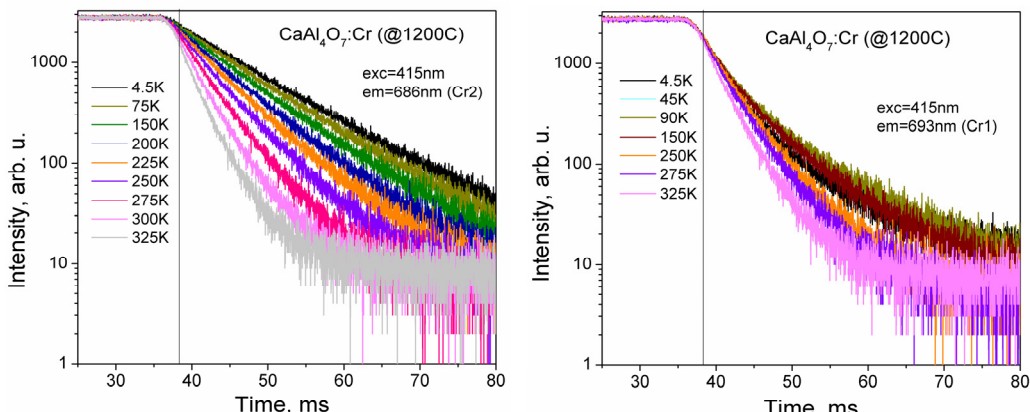

**Figure 10.** The temperature dependence of the luminescence decay curves of Cr1 and Cr2 centers in $CaAl_4O_7:Cr^{3+}$ calcined at 1200 °C.

It is noteworthy that the position of the R lines and their intensity ratio for the Cr1 center in the studied material are remarkably similar, if not identical, to those observed in ruby $\alpha$-$Al_2O_3:Cr^{3+}$ [17]. This similarity suggests that the Cr1 center may be related to $\alpha$-$Al_2O_3$, which is likely present as an unintentional impurity phase in the studied materials. However, XRD studies did not detect any traces of the $\alpha$-$Al_2O_3$ phase. Furthermore, a similar, albeit non-identical, "ruby-type" $Cr^{3+}$ center was also observed in isostructural $SrAl_4O_7$ (see Figure 6b). This suggests that $Cr^{3+}$ ions, when they are introduced to the $(Ca/Sr)Al_4O_7$ lattice, form octahedrally coordinated centers similar to those in ruby. It is logical to assume that this type of centers is formed in locations of $Al^{3+}$ ions. Due to the tetrahedral environment of $Al^{3+}$ ions in the $CaAl_4O_7$ structure, it can be assumed that a $Cr^{3+}$ ion, when it enters into an Al site, attracts two oxygens from the coordination environment of the two nearest calcium atoms, thus forming $CrO_6$ octahedra. A similar scenario of $CrO_6$ octahedral complex formation, i.e., the generation of distorted octahedra as $Cr^{3+}$ ions replaced $Ga^{3+}$ ions in an $SrGa_4O_7$ structure, was recently reported for isostructural $SrGa_4O_7$ in [18].

Another type of $Cr^{3+}$ center (Cr2), which showed up in the studied material at calcination temperatures $\leq 1200$ °C and is attributed to $Cr^{3+}$ in a somewhat weaker crystal field (larger Cr–O distances), is logically assigned to the centers formed at $Ca^{2+}$ ions sites. The presence of the corresponding Cr2 center, in a much weaker crystal field in $SrAl_4O_7$ as shown above, confirms this assignment. Note that a similar mechanism of the formation of distorted octahedral $Mn^{4+}$ complexes due to substitution for Ca sites was proposed in $CaAl_4O_7:Mn$ phosphors [8]. Owing to the peculiarity of the grossite structure, both scenarios of the formation of octahedral $Cr^{3+}$ centers in a $CaAl_4O_7$ host structure are possible. To shed light on this problem, a thorough study of the local structure of $Cr^{3+}$-doped $Ca(Sr)Al_4O_7$ materials via complementary techniques is required.

### 2.3. Evaluation of Feasibility of CaAl$_4$O$_7$:Cr$^{3+}$ for Non-Contact Luminescence Thermometry

The detailed investigation of changes observed in the spectroscopic characteristics of CaAl$_4$O$_7$:Cr$^{3+}$ with temperature indicates that alike of many other chromium-doped oxides, this material can be used as non-contact luminescence sensor for temperature monitoring. One key advantage of materials doped with ions of transition metals is the ability to employ various modalities for temperature sensing based on monitoring different temperature-dependent luminescence characteristics, such as the intensity ratio of R lines, their spectral shift, and decay time. All these characteristics exhibit notable changes with temperature [17], making multimodal non-contact luminescence measurements of temperature highly attractive. Multimodal sensing improves the accuracy and reliability of the technique by enabling the cross-referencing of results produced by different readout techniques. Recently, we demonstrated that further enhancement of the technique can be achieved by using two transition metal ions, Cr$^{3+}$ and Mn$^{4+}$, in an Al$_2$O$_3$ host [19]. However, the practical usefulness of this dual-emitter concept is limited due to the difficulty of accommodating ions with different valence states in one host. An alternative approach to this concept is to use one transition metal ion to form two types of emission centers with distinct spectral and decay characteristics. As is demonstrated above, CaAl$_4$O$_7$:Cr$^{3+}$, exhibiting the emissions of two types of Cr$^{3+}$-centers, is well suited to test this idea. This section analyzes the luminescence characteristics of CaAl$_4$O$_7$:Cr$^{3+}$ from the viewpoint of their application for thermometry.

The spectra presented in Figure 9 reveal significant changes in the intensity distribution of steep R lines and their shift with temperature. A rapid decrease in the intensity of the $R_2$ lines with respect to $R_1$ during cooling is due to an increase in the population of the lower exited state that is governed by Boltzmann statistics. The thermal shift of the lines results from the interaction of electronic states of impurity ions with acoustic phonons of the host matrix. That is explained within the framework of the Debye theory of solids. Consequently, both observed dependencies can be used for temperature monitoring.

The plots in Figure 11 show the variation in the ratio of intensities, $F = I_{R2}/I_{R1}$, of $R_2$ and $R_1$ lines with temperature, which is widely used in ratiometric optical thermometry. The experimental results were fitted by the equation that describes the temperature variation in the population of the emitting levels using Boltzmann statistics:

$$F(T) = A \exp(-\frac{D}{kT}), \tag{1}$$

where $A$ is a constant, $D$ is the activation energy for the transition between the two emitting levels, $k$ is the Boltzmann constant, and $T$ is the absolute temperature. As can be seen from the results of fitting the Boltzmann distribution, it effectively describes the observed temperature changes in the intensity ratio of $R$ lines emitted by both centers (Cr1 and Cr2) in CaAl$_4$O$_7$:Cr$^{3+}$. Deviations from the theoretical model observed at high temperatures (above 250 K) are due to the depopulation of emitting levels occurring via non-radiative decay processes. The values for the energy gap $D$ between the emitting levels, $\overline{E}$ and $\overline{2A}$, derived from the fittings are 4.7 $\pm$ 0.3 and 11.8 $\pm$ 0.4 meV for Cr1 and Cr2 centers, respectively. The larger energy gap obtained for Cr2 emission centers results in enhanced sensitivity of the ratiometric optical thermometer over a broader range of temperatures.

Figure 12 shows the temperature dependence of the spectral position of the $R_1$ and $R_2$ lines. The Debye theory gives the following expression to describe the thermal shift of the lines, $\Delta v$:

$$\Delta v(T) = \alpha (\frac{T}{T_D})^4 \int_0^{T_D/T} \frac{x^3}{\exp(x) - 1} dx \tag{2}$$

where $\alpha$ is the coupling coefficient for electron–phonon interactions, and $T_D$ is the Debye temperature. We performed a correlated fit of the observed temperature dependences of $R$ line positions, which allowed us to derive individual values for the parameter $\alpha$ of each line and the Debye temperature, $T_D$, common for both lines. The

fitting yielded $\alpha_{R1} = -531.1 \text{ cm}^{-1}$, $\alpha_{R2} = -515.4 \text{ cm}^{-1}$, and $T_D = 858$ K for Cr1 centers and $\alpha_{R1} = -379.3 \text{ cm}^{-1}$, $\alpha_{R2} = -498.7 \text{ cm}^{-1}$, and $T_D = 792$ K for Cr2 centers. From the data shown in Figure 12, it is obvious that at low temperatures, the emission peaks exhibit a small thermal shift. Measurable changes in the peak positions are observed as the temperature exceeds 100 K. This sets a natural low-temperature limit for the application of the method for non-contact temperature measurements.

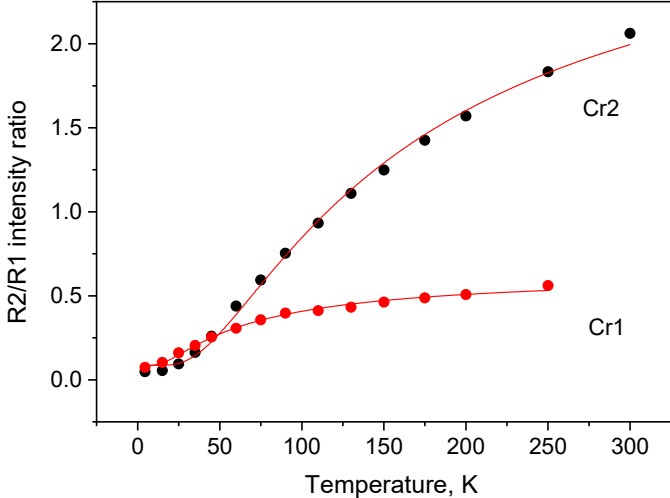

**Figure 11.** The luminescence intensity ratio of $R$ lines ($F = I_{R2}/I_{R1}$) emitted by Cr1 and Cr2 centers in $CaAl_4O_7:Cr^{3+}$ as a function of temperature. The red lines show the best fit of experimental results (dots) to Equation (1).

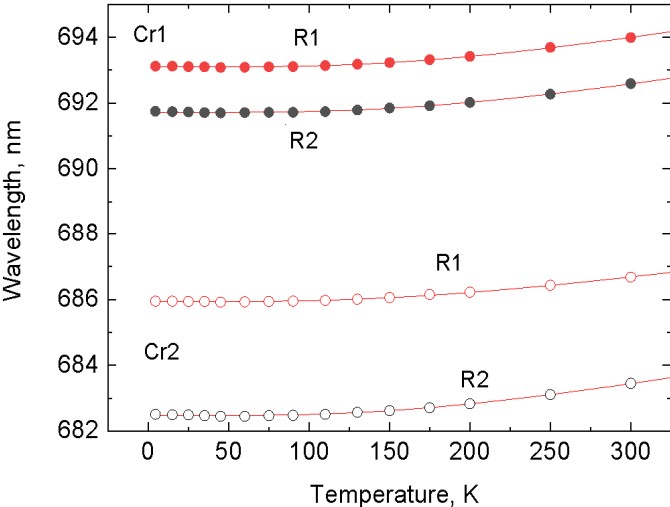

**Figure 12.** Temperature shift of the $R$ lines of Cr1 and Cr2 emission centers in $CaAl_4O_7:Cr^{3+}$. The dots are the experimental data points, and the solid lines show the best fit of the experimental results to Equation (2).

Complementary to the spectroscopic methods of non-contact temperature sensing is a technique that takes advantage of the changes of the decay rate of $Cr^{3+}$-emission with temperatures. The luminescence decay time constant of Cr2 center exhibits strong changes with temperature in contrast to the weak thermal dependence of the decay time constant observed for Cr1 center (refer to Figure 13). Furthermore, the decay rate of Cr1 centers is noticeably faster and not a single exponential function, indicating the presence of energy transfer processes. Consequently, only the temperature dependence of the luminescence decay of the Cr2 center is conducive for practical application in non-contact thermometry.

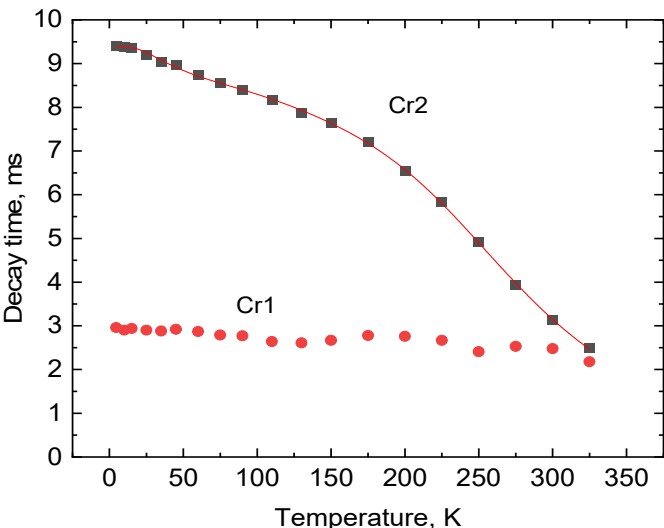

**Figure 13.** Temperature dependence of the luminescence decay time constant of Cr1 and Cr2 centers in CaAl$_4$O$_7$:Cr$^{3+}$. The solid lines show the best fit of Equation (3) to the experimental results (dots) using the following parameters: $\tau_1 = 9.33 \pm 0.02$ ms, $\tau_2 = 6.41 \pm 0.08$ ms, $\tau_3 = 0.067 \pm 0.013$ ms, $E_p = 46 \pm 1$ meV, $E = 152 \pm 5$ meV, and $D = 9$ meV.

The temperature dependences displayed in Figure 13 can be explained by considering the main processes affecting the population of emitting level $^2$E, including thermalization, phonon-assisted interaction with lattice vibrations, and thermally induced depopulation. Assuming that there is thermal equilibrium between states $\overline{E}$, $\overline{2A}$, and $^4$T$_2$ involved in the transitions and considering interactions with lattice vibrations, the expression for the temperature dependence of the decay time of Cr$^{3+}$ is given as follows [17]:

$$\tau(T) = \frac{1 + \exp(-\frac{D}{kT}) + \exp(-\frac{\Delta E_1}{kT})}{\frac{1}{\tau_1}\coth(\frac{E_p}{2kT}) + \frac{1}{\tau_2}\coth(\frac{E_p}{2kT})\exp(-\frac{D}{2kT}) + \frac{1}{\tau_3}\exp(-\frac{\Delta E_1}{kT})} \qquad (3)$$

where $1/\tau_i$ ($i = 1, 2$ and 3) is the radiative decay rates of the involved states, respectively, $k$ is the Boltzmann constant, $D$ is the energy split of the $^2$E levels, $\Delta E_1$ is the energy difference between the $^2$E and the upper state, and $E_p$ stands for "effective energy" of the phonons responsible for the exchange with sidebands. The correlation coefficient of the fit shown in Figure 13 is 0.999, evidencing that the used model perfectly represents the measured temperature dependences of the decay time constants of the Cr2 center.

Interestingly, the decay time of the lower emitting level, $\overline{E}$, is longer than that for $\overline{2A}$, and due to this, the heating results in a gradual increase in the overall transition rate from the two levels. This is in contrast to the situation in many other Cr-doped oxides, where a decrease in the rate observed during heating in the low-temperature range is due to the shorter decay time of the lower $\overline{E}$ level, which is depopulated through thermally activated transition to the upper $\overline{2A}$ level [17]. This relationship of two parameters ($\tau_1 > \tau_2$) has a quite important implication for temperature sensing, resulting in the monotonous decrease in the decay time constant and improved sensitivity over a broader range.

The discussion of the performance of the material for thermometry typically considers two parameters, i.e., absolute sensitivity, $S_a$, and relative sensitivity, $S_r$, which are defined as follows:

$$S_a = \left| \frac{dQ}{dt} \right| \qquad (4)$$

and

$$S_r = \left| \frac{dQ}{dt} \right| Q^{-1} \qquad (5)$$

where $Q$ is the absolute thermometric parameter of interest. The calculated values of these parameters are summarized in Figures S1–S4.

Finally, to compare the performances of different methods, we determined the temperature uncertainty of each method, $\delta T$, from the uncertainty of the measured parameter ($\delta Q$) using the equation and approach developed earlier in [20]:

$$\delta T = \delta Q \left| \frac{dQ}{dT} \right|^{-1} \tag{6}$$

where $|dQ/dT|$ is the absolute sensitivity of the thermometric parameter. The obtained characteristic can be readily used for the comparison of different temperature measurement methods. The plots in Figure 14 display the variation in uncertainty with temperature, $\delta T = f(T)$, calculated for different methods of non-contact sensing when $CaAl_4O_7:Cr^{3+}$ is used. The graphs illustrate that the accuracy of various measurement methods depends heavily on the temperature range. At low temperatures (10–140 K), measurements of the intensity ratio of emission peaks from the Cr2 center yields the lowest degree of uncertainty. As the temperature increases, more accurate measurements can be obtained by monitoring the changes in the luminescence decay time constant of Cr2 centers. In fact, using this method of temperature monitoring with $CaAl_4O_7:Cr^{3+}$ allows us to achieve a temperature resolution of more than $\pm 1$ K above 200 K. This performance is comparable to that of $Ga_2O_3$-Cr, which has been identified as one of the most sensitive Cr-doped compounds for non-contact luminescence thermometry in this temperature range [17]. Finally, it should be noted that the accuracy of temperature determination based on the shift of peak positions is much lower compared to those of other techniques.

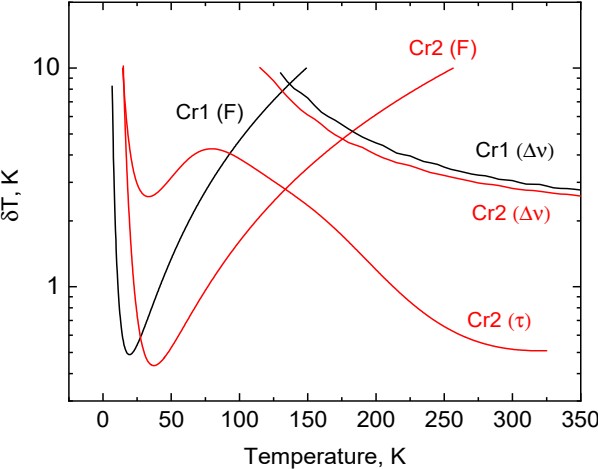

**Figure 14.** The uncertainty of temperature sensing of $CaAl_4O_7:Cr^{3+}$ using different modes of non-contact luminescence thermometry. *F*—luminescence intensity ratio; $\Delta v$—spectral shift; $\tau$—decay time constant.

## 3. Materials and Methods

### 3.1. Samples Preparation

A series of nanocrystalline $Cr^{3+}$-doped $CaAl_4O_7$ powders of nominal composition, $CaAl_{3.9995}Cr_{0.0005}O_7$, was obtained via a combined sol–gel solution combustion method, following by heat treatment at temperatures of 900 °C, 1000 °C, 1100 °C, 1200 °C, and 1300 °C. For this purpose, the following substances were used: $Ca(NO_3)_2 \cdot 4H_2O$ (Sfera Sim, Lviv, Ukraine), $Al(NO_3)_3 \cdot 9H_2O$ (Alfa Aesar, Haverhill, MA, USA), and $Cr(NO_3)_3 \cdot 9H_2O$ (Alfa Aesar, Haverhill, MA, USA), which were used as soluble salts of metal sources, citric acid $C_6H_8O_7$ (CA), and ethylene glycol $C_2H_6O_2$ (EG) (Sfera Sim, Lviv, Ukraine), which was used as a chelating agent and fuel. A total of 25 wt. % ammonia solution (Sfera Sim, Lviv, Ukraine) was used as a regulator of pH media (the concentrated solution

was used to minimize the resulting mixture volume). All substances were of an analytical grade of purity. The molar ratio of total metal ions, CA, and EG were kept as follows: $n(\Sigma Me^{n+}):n(CA):n(EG) = 1:2:8$. Each component was dissolved in a separate 100 mL flask, except for EG, to obtain a concentrated solution. Aliquots of each agent's solution were calculated according to their nominal composition, considering the desired mass of final products (typically from 0.75 to 2.0 g) and mixed in evaporating ceramic dishes with a volume of up to 200 mL. The obtained mixture of components was placed in an oven, pre-heated to $\approx$363 K for 15–30 min for it to completely dissolve, followed by neutralization of the resulting mixture to a pH equal to 7.0. After pH value correcting, to remove residual water and ensure the completeness of the esterification process, solutions were placed again in an oven heated to 398–423 K for 5–7 h. The process completion was monitored visually by the appearance of the gel, which changed from transparent to a brownish-black color. The obtained gel-like mixture of initial components was then cooled and placed into a muffle furnace preheated to 1073 K, which is necessary to initiate the component's decomposition and spontaneous combustion processes. Due to the rapid thermal decomposition and self-combustion of the reaction mixture, the formation of a bulky foamy product was observed. After furnace cooling, the obtained material was ground in a porcelain mortar, and subsequently, calcined in air at temperatures 900 °C, 1000 °C, 1100 °C, 1200 °C, and 1300 °C in order to reveal the impact of the heat treatment temperature on the crystal structure parameters, the average crystallite size of the powders, and luminescent characteristics of the prepared materials. The technological scheme of the sol–gel solution combustion synthesis of $Cr^{3+}$-doped $CaAl_4O_7$ is presented in Figure 15.

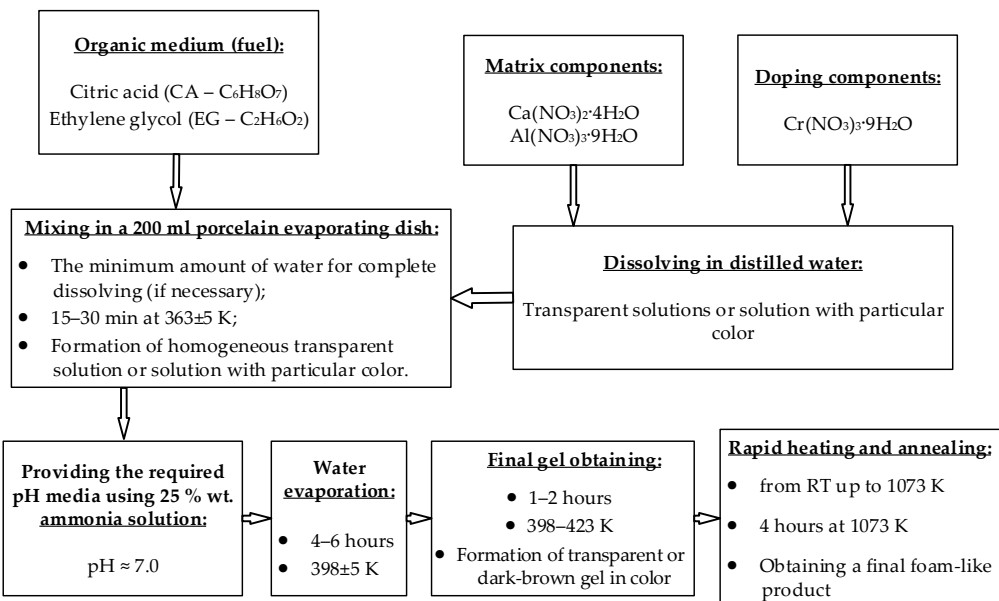

**Figure 15.** Technological scheme of sol–gel solution combustion synthesis of $Cr^{3+}$-doped $CaAl_4O_7$.

### 3.2. Physical–Chemical Characterization

The phase composition, crystal structure, and microstructural parameters of the materials were investigated via X-ray powder diffraction using the Aeris Research benchtop powder diffractometer (Malvern Panalytical, Almero, The Netherlands). XRD patterns were collected by analyzing Cu K$\alpha$ radiation ($\lambda$ = 1.54456 Å) in the 2$\theta$ range of 10–110 degrees, with a step of 0.02 deg. Analysis of the collected diffraction data was carried out using the WinCSD program package [21]. A full-profile Rietveld refinement technique was used for the evaluation of lattice parameters and positional and displacement parameters of atoms in $CaAl_4O_7$:$Cr^{3+}$ structures after heat treatment at different temperatures. The latest method was also used for the evaluation of average crystallite sizes and micro-strain values. To correct instrumental peak broadening, the $LaB_6$ external standard was used.

A Carl Zeiss AURIGA CrossBeam Workstation was used for the scanning electron microscope studies.

Photoluminescence and photoluminescence excitation spectra were measured using an Horiba/Jobin-Yvon Fluorolog-3 spectrofluorometer with a 450 W continuous spectrum xenon lamp for excitation and optical detection with a Hamamatsu R928P photomultiplier operating in photon counting mode. The measured photoluminescence excitation spectra were corrected with the emission spectrum from a xenon lamp. Photoluminescence spectra were corrected for the spectral response of the used spectrometer system. Luminescence decay kinetics were measured using the same Fluorolog-3 spectrofluorometer, with the excitation light modulated with a mechanical modulator (chopper). Spectroscopic measurements in the temperature range between about 4 and 325 K were acquired with a Janis continuous-flow liquid helium cryostat using a Lake Shore 331 temperature controller.

## 4. Conclusions

A series of $Cr^{3+}$-doped $CaAl_4O_7$ nanocrystalline powders with an average grain size of ~80 nm were obtained via a facile sol–gel combustion route, followed by successive heat treatments in air at 900, 1000, 1100, 1200, and 1300 °C. The formation of single-phase materials with a monoclinic grossite-type structure was proved via X-ray powder diffraction, which was supported by full-profile Rietveld refinement technique. Precise values of the lattice parameters, coordinates, and displacement parameters of atoms were derived for $CaAl_4O_7$:Cr samples annealed at the above five temperatures. A minor, but detectable, anisotropic effect of the annealing temperature on the unit cell dimensions of the $CaAl_4O_7$:Cr structure was found: a minor decrease in *a* and *b* lattice parameters, and the monoclinic angle *β* was accompanied by a small enhancement of the *c* parameter. A much more pronounced impact of the heat treatment temperature was revealed by studying the crystallinity of the samples, i.e., a detectable increase in the average grain size from 80 nm up to 170–190 nm, accompanied with a gradual decrease in micro strain values from 0.16% to 0.11% for samples annealed at 900 and 1300 °C, respectively. The peculiar part of the grossite structure is a presence of two non-equivalent tetrahedral positions of $Al^{3+}$ ions and one position of Ca ions in pentagonal bipyramidal coordination, all of which are, in principle, suitable for the incorporation of $Cr^{3+}$ ions into a $CaAl_4O_7$ lattice.

$CaAl_4O_7$:$Cr^{3+}$ powder exhibits NIR emission related to $^4T_2 \rightarrow ^4A_2$ and $^2E \rightarrow ^4A_2$ transitions in $Cr^{3+}$ ions. The temperature dependences of the *R* line positions, the luminescence intensity ratio of the *R* lines, and the decay time of $Cr^{3+}$ ions were studied. The luminescence intensity of $Cr^{3+}$ ions, which forms two types of centers in the $CaAl_4O_7$ structure, occupying sites in the aluminum (Cr1) and calcium (Cr2) octahedron, can be changed by amending the calcination temperature. Temperature-dependent luminescent spectra show different changes in the intensity of the $Cr^{3+}$ emission related to Cr1 and Cr2 centers in the temperature range 4.5–300 K. We evaluated the luminescence characteristics of $CaAl_4O_7$:$Cr^{3+}$ nanocrystalline phosphor to assess its potential for non-contact temperature monitoring. Our findings suggest that, such as other Cr-doped oxides, this material can be utilized via various modes of non-contact temperature sensing. It has been shown that the material enabled us to achieve a temperature resolution of more than ±1 K over a range of 200–300 K when using the luminescence decay time technique.

**Supplementary Materials:** The following supporting information can be downloaded at: https://www.mdpi.com/article/10.3390/inorganics11050205/s1.

**Author Contributions:** Conceptualization, L.V. and Y.Z.; validation, L.V. and A.S.; investigation, V.S., V.H., A.L. and Y.Z.; data analysis, V.M. and H.K.; writing—original draft preparation, L.V. and V.M.; writing—review and editing, L.V., V.M. and Y.Z.; visualization, V.S. and V.H.; supervision, L.V.; project administration, L.V.; funding acquisition, L.V. and Y.Z. All authors have read and agreed to the published version of the manuscript.

**Funding:** This research was funded by the National Research Foundation of Ukraine under grant no. 2020.02/0373 "Crystalline phosphors' engineering for biomedical applications, energy saving lighting and contactless thermometry" and by the Polish National Science Centre (project nos. 2018/31/B/ST8/00774 and 2021/40/Q/ST5/00336).

**Data Availability Statement:** Data are available on a request on corresponding author.

**Acknowledgments:** The work was funded by the National Research Foundation of Ukraine under grant no. 2020.02/0373 "Crystalline phosphors' engineering for biomedical applications, energy saving lighting and contactless thermometry" and by the Polish National Science Centre (project nos. 2018/31/B/ST8/00774 and 2021/40/Q/ST5/00336). The authors are grateful to Tomasz Wojciechowski from the Institute of Physics PAS for the SEM studies.

**Conflicts of Interest:** The authors declare no conflict of interest.

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
