# Peer review of "Synthesis, Crystal Structure and Photoluminescent Properties of Red-Emitting CaAl4O7:Cr3+ Nanocrystalline Phosphor"

_inorganics, doi:10.3390/inorganics11050205_

Round 1

Reviewer 1 Report

There are some observations:

1. Energy levels digram showing the transition involved would be useful.

2. The authors have to include their previous paper in the reference (and/or other ones about Cr3+ doped SrAl4O7 …or BaAl4O7 if exists…) and some comments about the possible influence of the alkaline-earth metal ion

3. A discussion/comments about the Cr3+ luminescence and strong or weak crystal-field in the phosphor according to the works of S. Adachi [Review—Photoluminescence Properties of Cr3+ - Activated Oxide Phosphors, 2021 ECS J. Solid State Sci. Technol. 10 026001; Luminescence spectroscopy of Cr3+ in an oxide: A strong or weak crystal-field phosphor? Journal of Luminescence 234 (2021) 117965] has to be included

Author Response

The authors are very grateful to the Reviewer for careful reading of the manuscript and valuable remarks made. Please find below our response on your suggestions:

  1. Energy levels digram showing the transition involved would be useful.

In our opinion, there is no need to show the energy level diagram as it is a canonical illustration that is also displayed in the quoted literature. The additional references on this issue were added to the revised version as suggested by the Reviewer (see comment 3 below).

  1. The authors have to include their previous paper in the reference (and/or other ones about Cr3+ doped SrAl4O7 …or BaAl4O7 if exists…) and some comments about the possible influence of the alkaline-earth metal ion

We would thanks the Reviewer for this suggestion. First, our previous paper on Cr3+ and Mn4+ doped SrAl4O7 is quoted here (Ref. [13]). To the best of our knowledge, there are no mentions in the literature of Cr3+ doped BaAl4O7. In SrAl4O7:Cr, previously we have not revealed a chromium center in a weaker crystal field similar to Cr2 center in CaAl4O7. However, having revisited raw spectra as well as new measurements performed for SrAl4O7:Cr samples we revealed a center in a much weaker crystal in SrAl4O7:Cr. These new data for SrAl4O7:Cr have been added to the revised version of the present manuscript as Fig. 7. Besides, we have added the comparison of the R-lines for the chromium center in a strong crystal field (a "ruby-type" Cr1 center) in SrAl4O7 and CaAl4O7 (inset in Fig. 6b). These results suggest that the alkaline-earth cation very slightly influences the formation of chromium centers in Al sites. Whereas the chromium center in a weaker crystal field (Cr2 center) strongly depends on the alkaline-earth cation (the crystal field strength in Sr- is much weaker than in the Ca-host that is governed by the ionic radii of the cations).

  1. A discussion/comments about the Cr3+ luminescence and strong or weak crystal-field in the phosphor according tothe works of S. Adachi [Review—Photoluminescence Properties of Cr3+ - Activated Oxide Phosphors, 2021 ECS J. Solid State Sci. Technol. 10 026001; Luminescence spectroscopy of Cr3+ in an oxide: A strong or weak crystal-field phosphor? Journal of Luminescence 234 (2021) 117965] has to be included

This has been addressed in the revised version of the manuscript. Suggested papers are added to the Reference list.

Reviewer 2 Report

The manuscript entitled “Synthesis, Crystal Structure and Photoluminescent Properties of Red-emitting CaAl4O7:Cr3+ Nanocrystalline Phosphor” was interesting and well-written by the authors. All the necessary characterizations were done and the discussion was supported by the experimental data. The authors clearly explained the structural properties and luminescence properties. Although the authors presented the decay curves of the material at different temperatures, they didn’t mention anything about the lifetime or quantum efficiency of the material. Also, the sensitivity calculation was a bit confusing and it would be better if the authors compared their work with the previous work and clearly mentioned the advantage of their material over it.  

Author Response

The authors are very grateful to the Reviewer for careful reading of the manuscript and valuable remarks made. Please find below our response on your report:

The manuscript entitled “Synthesis, Crystal Structure and Photoluminescent Properties of Red-emitting CaAl4O7:Cr3+ Nanocrystalline Phosphor” was interesting and well-written by the authors. All the necessary characterizations were done and the discussion was supported by the experimental data. The authors clearly explained the structural properties and luminescence properties.

Although the authors presented the decay curves of the material at different temperatures, they didn’t mention anything about the lifetime or quantum efficiency of the material. Also, the sensitivity calculation was a bit confusing and it would be better if the authors compared their work with the previous work and clearly mentioned the advantage of their material over it.   

The quantum efficiency has not been studied in this work.  The value of the decay time constant (=lifetime) is presented in Fig. 13 as a function of temperature. We added information on the absolute and relative sensitivity of different techniques in the Supplementary data file. The comparison with other, the best to the date material studied by us, has been clearly made in the following passage: This performance is comparable to that of Ga2O3-Cr, which has been identified as one of the most sensitive Cr-doped compounds for the non-contact luminescence thermometry in this temperature range.

Reviewer 3 Report

The manuscript "Synthesis, Crystal Structure and Photoluminescent Properties of Red-emitting CaAl4O7:Cr3+ Nanocrystalline Phosphor” deals with a series of calcium dialuminate CaAl4O7 nanopowders with grossite-type structure, doped with chromium ions, synthesized by the combined sol-gel solution combustion method. Evolution of phase composition, crystal structure and microstructural parameters of the nanocrystalline materials depending on the temperature of thermal treatment was investigated by X-ray powder diffraction, applying the Rietveld refinement technique. Photoluminescent properties of CaAl4O7 nanophosphors activated with Cr3+ ions were studied over the temperature range of 4.5-325 K. This paper reports some interesting findings about the structure-optical properties of phosphor materials. The authors have very nicely represented the work with some interesting discussions. I recommend this manuscript for publication. The manuscript, however, requires some mandatory revisions before being considered for publication.

1.     The introduction on recently developed photoluminescent materials needs to be further extended. This is a very fast-rowing research area and authors need to refer to papers that are more recent. I strongly advise authors consult following manuscripts:  

-        Kong, L., Sun, H., Nie, Y., Yan, Y., Wang, R., Ding, Q., Zhang, S., Yue, H., Luan, G. (2023). Luminescent Properties and Charge Compensator Effects of SrMo0.5W0.5O4:Eu3+ for White Light LEDs. Molecules, 28(6). doi: 10.3390/molecules28062681

-        Li, G., Huang, S., Li, K., Zhu, N., Zhao, B., Zhong, Q., Zhang, Z., Ge, D., Wang, D. (2022). Near-infrared responsive Z-scheme heterojunction with strong stability and ultra-high quantum efficiency constructed by lanthanide-doped glass. Applied Catalysis B: Environmental, 311, 121363. doi: https://doi.org/10.1016/j.apcatb.2022.121363

-        Jin, J.W., Lin, J., Huang, Y., Zhang, L., Jiang, Y., Tian, D., Lin, F., Wang, Y., Chen, X., High sensitivity ratiometric fluorescence temperature sensing using the microencapsulation of CsPbBr3 and K2SiF6:Mn-4(+) phosphor. CHINESE CHEMICAL LETTERS, 2022. 33(11): p. 4798-4802. 

2.     Shouldn’t “materials and methods” come before “results and discussions”?

3.     Can you further elaborate on the reason behind the shift in Figure 6a?

4.     Please index peaks in XRD graph.

5.     Is this such extent of scattering in data in Figure 9 to be expected?

6.     Conclusion seems to me a bit tedious and rather generic. Please re-write in bullet point and be more specific about your findings.  

Author Response

The authors are very grateful to the Reviewer for careful reading of the manuscript and valuable remarks made. Please find below our response on your report:

The manuscript "Synthesis, Crystal Structure and Photoluminescent Properties of Red-emitting CaAl4O7:Cr3+ Nanocrystalline Phosphor” deals with a series of calcium dialuminate CaAl4O7 nanopowders with grossite-type structure, doped with chromium ions, synthesized by the combined sol-gel solution combustion method. Evolution of phase composition, crystal structure and microstructural parameters of the nanocrystalline materials depending on the temperature of thermal treatment was investigated by X-ray powder diffraction, applying the Rietveld refinement technique. Photoluminescent properties of CaAl4O7 nanophosphors activated with Cr3+ ions were studied over the temperature range of 4.5-325 K. This paper reports some interesting findings about the structure-optical properties of phosphor materials. The authors have very nicely represented the work with some interesting discussions. I recommend this manuscript for publication. The manuscript, however, requires some mandatory revisions before being considered for publication.

  1. The introduction on recently developed photoluminescent materials needs to be further extended. This is a very fast-rowing research area and authors need to refer to papers that are more recent. I strongly advise authors consult following manuscripts:  

-        Kong, L., Sun, H., Nie, Y., Yan, Y., Wang, R., Ding, Q., Zhang, S., Yue, H., Luan, G. (2023). Luminescent Properties and Charge Compensator Effects of SrMo0.5W0.5O4:Eu3+ for White Light LEDs. Molecules, 28(6). doi: 10.3390/molecules28062681

-        Li, G., Huang, S., Li, K., Zhu, N., Zhao, B., Zhong, Q., Zhang, Z., Ge, D., Wang, D. (2022). Near-infrared responsive Z-scheme heterojunction with strong stability and ultra-high quantum efficiency constructed by lanthanide-doped glass. Applied Catalysis B: Environmental, 311, 121363. doi: https://doi.org/10.1016/j.apcatb.2022.121363

 -        Jin, J.W., Lin, J., Huang, Y., Zhang, L., Jiang, Y., Tian, D., Lin, F., Wang, Y., Chen, X., High sensitivity ratiometric fluorescence temperature sensing using the microencapsulation of CsPbBr3 and K2SiF6:Mn-4(+) phosphor. CHINESE CHEMICAL LETTERS, 2022. 33(11): p. 4798-4802. 

The statement about the activity in the field of photoluminescent material is generally correct however we are puzzled how this can be linked with the suggestion of referee to quote this particular selection of papers. They deals with very different types of materials, i.e. mixed tungstates-molibdates, halides and are not relevant to justify quotation in the staudy on luminescence of the Cr-doped oxide.

  1. Shouldn’t “materials and methods” come before “results and discussions”?

According to a template for manuscript preparation, the “Materials and Methods” section should come after the “Results” and “Discussions” sections.

  1. Can you further elaborate on the reason behind the shift in Figure 6a?

The red shift of the excitation bands (4A24T1 and 4A24T2 transitions in the octahedrally coordinated Cr3+ ion) for Cr2 centers with respect to Cr1 centers is due to a lower strength of crystal field experienced by Cr2 in comparison with Cr1 centers.

  1. Please index peaks in XRD graph.

The main Millers indices are added to the XRD graph on Figure 1.  

  1. Is this such extent of scattering in data in Figure 9 to be expected?

Affirmative, this is a typical appearance of experimentally measured decay curves presented in a log scale.

  1. Conclusion seems to me a bit tedious and rather generic. Please re-write in bullet point and be more specific about your findings.

Many thanks for the advice but we disagree with the concept of “bullet points” conclusions that are more appropriate style for highlights. To our opinion conclusion should be written exactly as it is - a proper summary of the work undertaken and its findings. No changes here.

Reviewer 4 Report

The article is devoted to the study of the properties of CaAl4Onanophosphors activated with Cr3+ ions, which can potentially be used for non-contact luminescent thermometry in the temperature range 4.5-325 K. This is a very hot topic in materials science and inorganic chemistry, therefore, I am sure that the article will arouse interest among scientists involved in developments in this area.

The manuscript is well written and structured, the data is presented logically and the general essence of the article is easily captured. In general, after minor modifications, the manuscript can be recommended for publication in INORGANICS.

1.      It is not at all clear from the text of the article what proportion of Cr3+ ions was chosen for doping and what determines the proportion of doping ions that was ultimately chosen for the study. Perhaps it is worth duplicating this information from the experimental part and giving some explanations.

2.      The article provides data on particle size, but, as far as it follows from the manuscript, it was calculated from X-ray data. Additional particle size studies, such as Dinamic Light Scattering or microscopy, should be given.

3.      Formulae and some other designations in the text are inserted with pictures, it is worth replacing with text to facilitate the perception of the article.

4.      The values of absolute and relative temperature sensitivity, which are important characteristics of fluorescent thermometers, are not given.

5.      There is no data on the repeatability of the obtained temperature dependence.

Author Response

The authors are very grateful to the Reviewer for careful reading of the manuscript and valuable remarks made. Please find below our response on your remarks:

The article is devoted to the study of the properties of CaAl4O7 nanophosphors activated with Cr3+ ions, which can potentially be used for non-contact luminescent thermometry in the temperature range 4.5-325 K. This is a very hot topic in materials science and inorganic chemistry, therefore, I am sure that the article will arouse interest among scientists involved in developments in this area.

The manuscript is well written and structured, the data is presented logically and the general essence of the article is easily captured. In general, after minor modifications, the manuscript can be recommended for publication in INORGANICS.

  1. It is not at all clear from the text of the article what proportion of Cr3+ions was chosen for doping and what determines the proportion of doping ions that was ultimately chosen for the study. Perhaps it is worth duplicating this information from the experimental part and giving some explanations.

The concentration of Cr3+ ions in the studied material is determined by the nominal composition of the compound as CaAl3.9995Cr0.0005O7. Corresponding clarifications have been added to the revised manuscript. Such proportion of activator was chosen based on our own and the literature data on Cr-doped alumina-based materials.   

  1. The article provides data on particle size, but, as far as it follows from the manuscript, it was calculated from X-ray data. Additional particle size studies, such as Dinamic Light Scatteringor microscopy, should be given.

Following the Reviewers recommendation, we added to the manuscript SEM pictures of the studied samples, annealed at different temperatures.

  1. Formulae and some other designations in the text are inserted with pictures, it is worth replacing with text to facilitate the perception of the article.

This has been corrected.

  1. The values of absolute and relative temperature sensitivity, which are important characteristics of fluorescent thermometers, are not given.

The absolute and relative sensitivities that can be achieved by different methods of temperature sensing have been included in the Supplementary data file.

  1. There is no data on the repeatability of the obtained temperature dependence.

The reproducibility of the temperature values obtained using non-contact techniques is a complex parameter that depends on both the reproducibility of the set temperature value and the precision of temperature determination. While as a technical characteristic this is relevant to our opinion there may not be much new scientific information to be gained from such a study in addition to what has already been discussed. Therefore, we did not conduct such examinations.

Round 2

Reviewer 1 Report

The authors provided the proper answers to the comments.

Reviewer 3 Report

The revision is acceptable. Thanks for inviting me for the review.